# Efficient Large-Scale GPS Trajectory Compression on Spark: A Pipeline-Based Approach

**Wen Xiong** [1,2], **Xiaoxuan Wang** [1,2,*] **and Hao Li** [1]

1   School of Information, Yunnan Normal University, Kunming 650500, China; wen.xiong@ynnu.edu.cn (W.X.); hao_li0520@163.com (H.L.)
2   Engineering Research Center of Computer Vision and Intelligent Control Technology, Yunnan Provincial Department of Education, Kunming 650500, China
*   Correspondence: wangxiaoxuan@ynnu.edu.cn

**Abstract:** Every day, hundreds of thousands of vehicles, including buses, taxis, and ride-hailing cars, continuously generate GPS positioning records. Simultaneously, the traffic big data platform of urban transportation systems has already collected a large amount of GPS trajectory datasets. These incremental and historical GPS datasets require more and more storage space, placing unprecedented cost pressure on the big data platform. Therefore, it is imperative to efficiently compress these large-scale GPS trajectory datasets, saving storage cost and subsequent computing cost. However, a set of classical trajectory compression algorithms can only be executed in a single-threaded manner and are limited to running in a single-node environment. Therefore, these trajectory compression algorithms are insufficient to compress this incremental data, which often amounts to hundreds of gigabytes, within an acceptable time frame. This paper utilizes Spark, a popular big data processing engine, to parallelize a set of classical trajectory compression algorithms. These algorithms consist of the DP (Douglas–Peucker), the TD-TR (Top-Down Time-Ratio), the SW (Sliding Window), SQUISH (Spatial Quality Simplification Heuristic), and the V-DP (Velocity-Aware Douglas–Peucker). We systematically evaluate these parallelized algorithms on a very large GPS trajectory dataset, which contains 117.5 GB of data produced by 20,000 taxis. The experimental results show that: (1) It takes only 438 s to compress this dataset in a Spark cluster with 14 nodes; (2) These parallelized algorithms can save an average of 26% on storage cost, and up to 40%. In addition, we design and implement a pipeline-based solution that automatically performs preprocessing and compression for continuous GPS trajectories on the Spark platform.

**Keywords:** trajectory compression; big data; spark; parallelized algorithm



## 1. Introduction

With the development of urban public transportation systems, the number of vehicles equipped with GPS (global positioning system) positioning devices is increasing. Every day, hundreds of thousands of vehicles, including buses, taxis, and ride-hailing cars, generate a large amount of GPS trajectory datasets. This massive GPS trajectory dataset contains a lot of valuable spatial and temporal characteristics information, which forms the foundation for a wide range of smart city applications. These applications include urban planning, route network construction, vehicle scheduling, emergency management, and public services [1]. Therefore, it is imperative to preserve this valuable trajectory dataset for future downstream applications.

However, these incremental and historical trajectory datasets require a lot of storage and computing resources, which brings unprecedented pressure to the owners of the traffic big data platform. First, the massive GPS trajectory data requires a large amount of storage space. Second, the massive GPS trajectory data requires many computing resources when the owners perform data analysis and mining tasks. Third, the massive

amount of data renders traditional visualization methods ineffective [2]. In order to alleviate the above pressure, GPS trajectories need to be compressed. If we can reduce the data volume of the original trajectory dataset, we can consequently reduce the costs associated with subsequent data storage, movement, computation, and visualization. Many researchers have explored trajectory compression algorithms, and all of which work well with small data sizes. Currently, these classical GPS trajectory compression algorithms include DP (Douglas–Peucker) [3], TD-TR (Top-Down Time-Ratio) [4], SW (Sliding Window) [5], SQUISH (Spatial Quality Simplification Heuristic) [6], and V-DP (Velocity-Aware Douglas–Peucker) [7].

The first purpose of GPS trajectory compression is to eliminate the redundant points that contain less information from the original trajectory while ensuring the accuracy of vehicle trajectory. The second purpose is to minimize the amount of trajectory data while satisfying the similarity conditions between the original trajectory and the compressed trajectory. The compressed trajectory, compared with the original data, will bring a great improvement to data mining speed due to the reduction in data volume. For example, if we save the compressed trajectories in a spatial-temporal database, the size of the compressed table will be significantly smaller than that of the original table, as well as its corresponding index. That benefits a set of spatial-temporal queries, such as the GPS KNN (k-nearest neighbors) query and the range query. Currently, trajectory compression research can be discussed in two dimensions [1]. First, the type of GPS trajectory, GPS trajectory consists of the trajectory of ground vehicles and the trajectory of AIS (automatic identification system) vessels in the ocean or river; second, the compression scenario, compression scenario consists of the offline and online manners. It should be especially noted that this study is designed to solve the problem of compressing large-scale vehicle GPS trajectories in an offline environment.

In the case of massive incremental and historical trajectory data, these algorithms are insufficient to compress an incremental GPS dataset with hundreds of GB in an acceptable time cost. The reason is that these algorithms can only be executed in a single-threaded manner and are limited to running in a single-node environment. Some researchers have used the MapReduce [8] framework in the Hadoop platform to compress GPS trajectory data and have also achieved good compression efficiency [9–11]. However, the efficiency and scalability of trajectory compression face challenges in further improvement due to the inherent limitations of the MapReduce programming model.

The emergence of Spark [12] makes it possible to further improve the efficiency of large-scale GPS trajectory compression. In detail, the Spark programming model differs from the MapReduce programming model in two aspects. On one hand, the former is a disk-based computation engine that requires data exchange between jobs and tasks through disks as the medium, leading to time-consuming I/O operations. The latter provides in-memory data exchange between jobs and tasks with the help of Resilient Distributed Dataset (RDD), which is a memory-based media [13]. On the other hand, the MapReduce programming model has only two operators, *Map* and *Reduce*, through which all complex computational logic can be expressed. Spark implements a DAG (directed acyclic graph)-based programming model, which provides more than twenty common operators and is much more expressive.

Therefore, Spark has an inherent advantage over MapReduce in terms of its design mechanism. In this paper, we try to parallelize these different classical compression algorithms in Spark. We then systematically evaluate these parallelized algorithms based on a large-scale real GPS trajectory dataset. Firstly, the experimental data is preprocessed in ETL (extract-transform-load) stage; secondly, we conduct the map matching algorithm [14] on the original GPS trajectory dataset to correct the position errors. Thus, all trajectory points are matched to the road network; finally, we design a spatial-temporal *partitioner* for RDD to dispatch massive trajectory to different nodes and perform trajectory compression using a group of tasks in a parallel way.

In summary, our contributions are as follows:

(1) We design and implement a pipeline-based solution that automatically performs pre-processing and compression for continuous GPS trajectories on the Spark platform. All data-processing steps, such as noise filtering, map matching, data partitioning, and trajectory compression, can be implemented as user-defined functions. This allows the operator to finely customize and control the data-processing workflow according to specific requirements.

(2) We parallelize a set of classical algorithms in Spark to meet the requirements of compressing large-scale GPS trajectory dataset. These algorithms consist of the DP, TD-TR, SW, SQUISH, and V-DP. All parallelized algorithms are seamlessly integrated into the pipeline environment along with data pre-processing steps.

(3) We systematically evaluate these classical algorithms on a very large dataset using different performance metrics. The experimental results show that: (1) It only takes 438 s to compress 117.5 GB GPS trajectory data on a 14-node Spark cluster; (2) these parallelized algorithms can save 26% storage cost on average, and up to 40% storage cost.

This paper is organized as follows: Section 1 provides an overview of our research; Section 2 shows the motivation and background; Section 3 discusses the related work; Sections 4 and 5 present the problem definition and the research methodology, respectively; Section 6 describes the experimental results and analysis; Section 7 provides a summary of our research.

## 2. Motivation and Background

### 2.1. A Big Data Platform for Transportation System

In order to safely and efficiently manage the modern public transport system, the city managers have built a traffic big data platform as an infrastructure. As shown in Figure 1, the platform is divided into three layers, which are the bottom layer, data warehouse layer, and application layer, respectively. The bottom layer is the data collection layer, which has built multiple sensor networks in the past years. Each sensor network contains thousands of terminals deployed on vehicles or stations, responsible for collecting one or more types of data. Currently, these types include GPS trajectory dataset, smart card dataset, and vehicle scheduling. The second layer is the data warehouse layer, in which it reorganizes the original heterogeneous datasets to different subject databases. The top layer is the application layer, which supports a wide range of applications. In addition, these applications can be classified into two categories, decision-making and public service. All applications adopt machine learning, data mining, and other algorithms to support decision-making or public service. For example, applications for decision making include vehicle scheduling and risk management, applications for public service include travel planning and route choice.

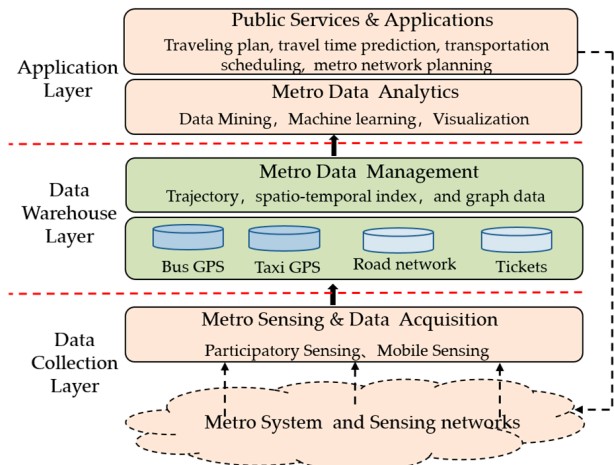

**Figure 1.** Transportation big data platform.

*2.2. GPS Trajectory and Storage Requirement*

Taking the public transportation system in Shenzhen as an example, it includes 19,000 buses, 20,000 taxis, and 80,000 ride-hailing cars. Every day, these vehicles run on the road network and provide services to millions of urban residents, which continuously produce GPS trajectory data. Currently, the big data platform has already collected a huge amount of GPS trajectory datasets while the incremental GPS data has been consistently increasing.

The incremental and historical GPS data require more and more storage space, placing unprecedented pressure on the data owners of the traffic big data platform. Assuming each vehicle generates one GPS record every 30 s, and each record contains 200 bytes, every vehicle generates 28,800, 86,400, and 1.0368 million records per day, month, and year, respectively. All vehicles generate 124.416 billion GPS records annually, resulting in a corresponding data size of 2488.32GB, consuming three times the storage space as data replication in HDFS. The cumulative data volume for five years is 36 TB, with incremental data amounting to 100 GB per day.

Figure 2 displays the time consumption of five different compression algorithms when compressing 1 GB trajectory data in a single-node environment. The TD-TR algorithm exhibits the longest time, which takes 320 s, while the SQ algorithm takes the shortest time, taking 119 s. The average execution time for the five methods is 182 s. It means the average time cost would exceed 5 h if one of these algorithms is used to compress 100 GB of GPS trajectory dataset. We can conclude that the algorithm executed in a single-threaded manner running on a single-node cannot meet efficiency requirement for the subsequent applications.

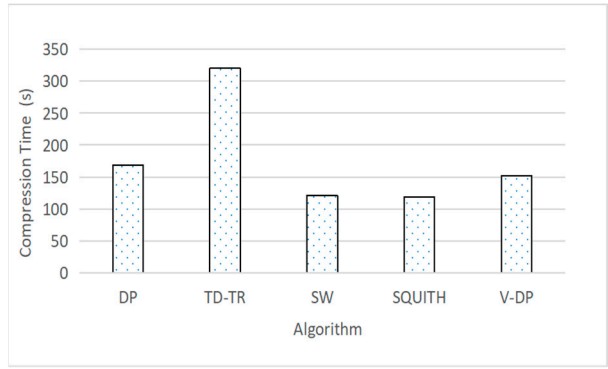

**Figure 2.** Compression times of different algorithms with 1 GB data in single node.

To address this deficiency, this paper endeavors to parallelize a set of classic compression algorithms using the big data computing engine Spark. By doing so, it aims to offer ample computational capabilities to fulfill the requirements of large-scale trajectory compression and achieve the objective of reducing storage costs.

## 3. Related Work

GPS trajectory compression is a classic research problem, and it is an issue of sustainability. Currently, many studies have been carried out on GPS trajectory compression. This section will first introduce the classic trajectory compression algorithms and their corresponding derivatives, and then discuss how big data technologies support trajectory compression scenarios. Ref. [15] conducted a comprehensive comparison of state-of-the-art trajectory simplification algorithms and assessed their quality using datasets that capture a range of diverse motion patterns.

The Bellman algorithm proposed in 1961 [16] is considered the first algorithm for trajectory simplification. The Bellman algorithm uses the dynamic programming method to find subsequence with N points, which makes the line segment with these points closest to the original curve, then greatly minimizes the space distance error between the compressed

trajectory and the original trajectory. But this algorithm has a huge cost, and its time complexity is O $(n^3)$. The DP algorithm proposed by Douglas and Peucker in 1973 has already became one of the most classic trajectory compression algorithms [4]. The DP algorithm is widely used due to its simple and efficient compression technology, but it is only suitable for batch processing and has significant compression errors. This algorithm considers spatial characteristics but ignores time characteristics.

In 2004, Meratnia et al. proposed a top-down time proportional trajectory compression algorithm called TD-TR [4]. TD-TR improved the DP algorithm by taking velocity and temporal characteristics into consideration. In the distance calculation process, synchronous Euclidean distance was used to replace perpendicular Euclidean distance. Jonathan Muckell et al., proposed the SQUISH algorithm in 2011 [6]. This algorithm deletes unimportant trajectory points in the buffer, updates the priority of the trajectory points, that is, using the priority of the deleted trajectory points to cover the priority of the adjacent trajectory points to protect the information of trajectory points adjacent to the deleted point. The algorithm has a time complexity of O (N × logβ) and a space complexity of O (β). Where N represents the number of trajectory points and β indicates the size of the buffer.

Some studies have extended these classical GPS trajectory compression algorithms [17–19] by considering the vehicle direction, speed, trajectory shape, and other factors. Some studies considered the road network [20–24], where the GPS trajectory is located, converted the original GPS trajectory into segment sequences, and then compressed the segment sequences. In addition to the compression of vehicle GPS trajectories, some studies explored the compression problem [25–29] for ship GPS trajectories collected by AIS. Beyond GPS trajectory compression in offline scenarios, some studies have also delved into real-time GPS trajectory compression algorithms. Ref. [4] proposed the Opening Window algorithm (OPW), which is the first online trajectory compression algorithm. TrajCompressor [30] have proposed an online compression algorithm; first conducting map matching of GPS and then considering vehicle direction and angle changes.

The above algorithms have achieved good compression results in their respective scenarios and can meet the data compression requirements of small-scale datasets. With the continuous increase in trajectory data size, some researchers are beginning to resort to big data technologies to address the large-scale GPS trajectory compression. Refs. [9,10,12] used Hadoop/MapReduce, a big data computing engine, to conduct massively parallel compressing for large-scale GPS trajectory dataset. Ref. [29] used GPU for GPS trajectory compression, which has a significant performance improvement compared to the CPU-based methods. Ref. [31] employed OpenMP [32] and GPU to explore online trajectory compression algorithm. It can satisfy the requirement of processing large-scale dataset in real-time. It would be an optimal solution if it could implement a task scheduler among different GPU or nodes. However, because of the natural limitations of the MapReduce and OpenMPI programming models, further improvements in efficiency and scalability are challenging.

Besides the advantages discussed in Section 1, Spark provides a powerful user-defined function mechanism, enabling end users to accomplish a series of independent data-processing steps. For instance, all data-processing steps, such as noise filtering, map matching, data partitioning, and trajectory compression, can be implemented as user-defined functions. This capability allows operators to finely customize and control the data-processing workflow according to their specific requirements. In contrast, other parallel computing frameworks, such as MapReduce and OpenMPI, lack such capability. For instance, managing data-processing steps becomes challenging when all steps are confined within a single pair of Map and Reduce functions. Similarly, the efficiency of data exchange among different jobs diminishes if each step is implemented within an isolated job.

In this study, we attempt to adopt Spark, which is a more advanced big data engine than MapReduce, to parallelize trajectory compression algorithms, expecting better efficiency and scalability.

## 4. Problem Definition

**Definition 1**. *Point.*

A point consists of a geographic coordinate and a timestamp. The coordinate is a two-dimensional tuple, e.g., (*latitude*, *longitude*). The timestamp records the time when the coordinate is read by a GPS positioning device. As shown in Equation (1):

$$P = (t, lat, lon) \tag{1}$$

In Equation (1), $t$ represents the timestamp of point $P$ and (*lat*, *lon*) represents the geographic coordinates. Thus, a point is a tuple with three attributes.

**Definition 2**. *Trajectory.*

A GPS trajectory consists of an identity and a sequence of points. The identity is the unique identity of a vehicle. All points in the sequence are ordered in time dimension. We denote $T$ as GPS trajectory and $T$ is shown in Equation (2):

$$T = (ID, P_1, P_2, \cdots, P_n) \tag{2}$$

In Equation (2), *ID* is the unique identity of a vehicle, and $n$ is the number of points and the length of trajectory $T$.

**Definition 3**. *Perpendicular Euclidean Distance (PED).*

The Perpendicular Euclidean distance between the trajectory point $P_m(x_m, y_m)$ and trajectory segment $\overline{P_s P_e}$ is the shortest distance from $P_m$ to $\overline{P_s P_e}$. The PED is defined in Equation (3):

$$PED(P_m) = \frac{|(y_e - y_s)x_m - (x_e - x_s)y_m + x_e y_s - y_e x_s|}{\sqrt{(y_e - y_s)^2 + (x_e - x_s)^2}} \tag{3}$$

As shown in Figure 3, assuming that trajectory points $P_1$ and $P_2$ are retained when $P_3$ is discarded, $P_3\prime$ is the projection point of $P_3$, which is located in trajectory segment $\overline{P_1 P_2}$. According to the definition of Perpendicular Euclidean distance, PED ($P_3$) is the shortest distance from point $P_3$ to segment $\overline{P_1 P_2}$.

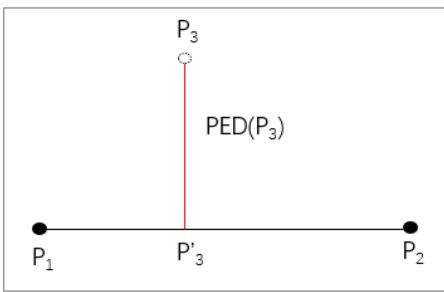

**Figure 3.** Perpendicular Euclidean distance.

**Definition 4**. *Synchronized Euclidean Distance (SED).*

When compared to Perpendicular Euclidean distance, Synchronized Euclidean Distance consider the time dimension, it assumes the speed between two points is a constant. Assuming that the point on the trajectory segment $\overline{P_s P_e}$, named $P'_m(t'_m, x'_m, y'_m)$, is the syn-

chronization point of trajectory point $P'_m$, then the synchronized Euclidean distance from $P_m$ to $P'_m(t'_m, x'_m, y'_m)$ is defined in Equation (4):

$$SED(P_m) = \sqrt{(x_m - x'_m)^2 + (y_m - y'_m)^2} \tag{4}$$

$$x'_m = x_s + \frac{x_e - x_s}{t_e - t_s}(t_m - t_s) \tag{5}$$

$$y'_m = y_s + \frac{y_e - y_s}{t_e - t_s}(t_m - t_s) \tag{6}$$

In these equations, $x'_m$ in Equation (5) and $y'_m$ in Equation (6) are synchronization points.

As shown in Figure 4, assuming that trajectory points $P_1$ and $P_2$ are retained while $P_3$ is discarded, $P_3$ is the synchronization point of $P_3$, which is located in trajectory segment $\overline{P_1 P_2}$.

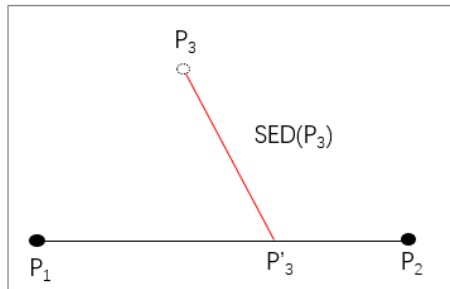

**Figure 4.** Synchronous Euclidean distance.

## 5. Method Design

Figure 5 fully describes the technology roadmap in this paper. The main steps include data acquisition, data pre-processing, algorithm parallelization, and evaluation.

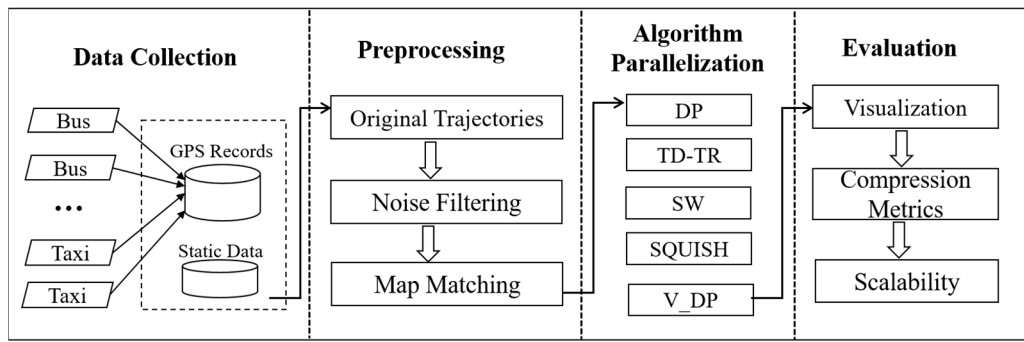

**Figure 5.** Technology roadmap.

### 5.1. Data Pre-Processing

GPS positioning devices deployed on buses and taxis read and transmit GPS records to the big data platform through mobile Internet or 4G/5G. The big data platform receives and retains these GPS trajectory datasets. Besides GPS data, the platform also stores other data sets, for example, bus lines and stations, bus scheduling, road network, smart card records.

Due to the limitations of the positioning device's accuracy, signal instability, and other reasons, various data quality issues exist with the original GPS trajectory data. For example, there are many outliers in GPS points, and many GPS points deviate from the actual road network. Before trajectory compression, we conduct data pre-processing on the original GPS dataset. The pre-processing mainly includes two parts, GPS noise filtering and GPS map matching.

### 5.1.1. GPS Noise Cancellation

According to data exploration, abnormalities in GPS trajectories can be divided into three categories: range anomaly, jump anomaly, and position coincidence. Generally, a vehicle's trajectory should be located on continuous road segments, and the corresponding sequence of GPS points should fall on the road network in the digital map. The details of these three categories are as follows:

- Range anomaly. The normal longitude range is 0–180°, and the normal latitude range is 0–90°. A range anomaly point means its longitude or latitude is not within the normal range. Such abnormal points are usually generated by errors in the positioning devices.
- Jump anomaly. Although the longitude and latitude of GPS are both within the normal range, obvious jump points still appear. It means a point significantly deviates from the rest points of the original trajectory. Possible reasons for the deviation include the accuracy of the positioning device and the obstruction of GPS signals by high-rise buildings.
- Position coincidence. This kind of point is a sequence of continuous GPS points, characterized by different time stamps, with a non-zero speed but no changed position. These kinds of anomalies are often caused by tunnels or viaducts. In detail, the GPS signal of a vehicle is being blocked and its location cannot be updated on time when passing through the tunnel or under the viaduct.

This paper utilizes an algorithm based on the heuristic GPS exception filtering [1]. The core idea of this algorithm is to calculate the difference between adjacent points in distance and time to obtain the instantaneous speed of the bus, and to determine whether it is an abnormal point by comparing the obtained speed with the specified speed threshold. During the detection process, we recorded the weights of abnormal data. Finally, abnormal points with high weight values will be removed from the original trajectory. After that, the filtered trajectory will be returned.

### 5.1.2. GPS Map Matching

A set of algorithms, including FMM [14] and ST-Matching [33], can address map matching for raw GPS trajectories. Based on the evaluation results in Ref. [14], this paper has employed the Fast Map Matching Algorithm (FMM) to rectify positioning errors in the original GPS trajectory.

FMM combines HMM (Hidden Markov Model) and pre-computation technology, which takes an original trajectory, road network as input, and outputs the matched GPS trajectory. After, a group of points that deviate from the road network are corrected, and all points of the matched GPS trajectory are located in the road network. Assume the input trajectory is represented as *TR*, and the road network graph is denoted *G<V, E>*. In detail, *E* is a set of edges, each edge stands for a road segment. *V* is a set of vertices, a pair of one start vertex, one end vertex forms an edge. For each trajectory, it will export a path and store this path as sequence of edges and their corresponding geometries. Using an Upper Bound Origin Destination Table (UBODT), pre-computation stores all road network shortest path pairs within a certain threshold.

In the first stage, the pre-computation algorithm takes the road network graph and the distance upper bound Δ of all shortest path pairs as inputs, then calculates the single-source shortest path and outputs the Upper Bound Origin Destination Table. In the second stage, based on considering the GPS error and topological constraint, HMM is incorporated with pre-computation to infer the path that the vehicle passed through. In detail, this stage can be further divided into four steps: CS (Candidate search), OPI (OPI integrated with UBODT), CPC (complete path construction), and GC (geometry construction). The CS step searches for the corresponding candidate edges for each point in the trajectory. Based on the HMM model, the OPI step firstly constructs a transition graph of candidate trajectories and queries the SP (shortest path pair) distance among candidate trajectories. Then, it derives the optimal path of the trajectory. In the CPC step, the SPs of continuous candidate paths in the optimal path will be connected to construct a complete path. The GC step

constructs the corresponding geometric. Finally, after the above processes, the original GPS trajectory can be corrected onto the road network of the digital map.

Figure 6 shows the comparison between an original trajectory (left side) and the corresponding matched trajectory (right side). We can clearly see from this figure that a group of green circles deviates from the road segments, while all points in the matched trajectory are located on the road segments.

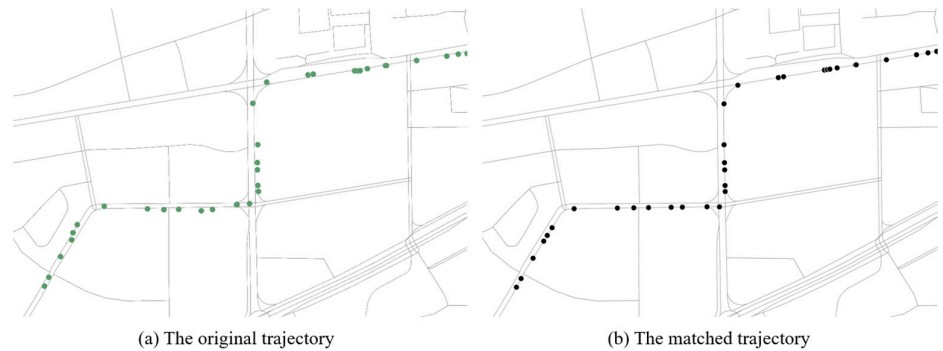

(a) The original trajectory                                          (b) The matched trajectory

**Figure 6.** The comparison between the original trajectory (**a**) and the matched trajectory (**b**).

### 5.2. Trajectory Compression Method

This paper has parallelized five classic trajectory compression algorithms in Spark platform. These algorithms are DP algorithm, SW algorithm, TD-TR algorithm, SQUISH, and V-DP algorithm. Due to space limitation, this section only provides detailed descriptions of DP and SW algorithms, details about TD-TR algorithm, SQUISH algorithm, and V-DP algorithm can be found in Refs. [4,6,7], respectively. We have referenced the approach outlined in Ref. [7] and made improvements to it. Therefore, we call the improved algorithm V-DP (Velocity-Aware Douglas–Peucker).

#### 5.2.1. DP Algorithm

The DP algorithm is one of the most classic offline compression algorithms. The principle is to reduce the number of points in the original trajectory, and make the compressed trajectory approximate the original trajectory while ensuring compression ratio. The distance used in the DP algorithm is the perpendicular Euclidean distance. Figure 7 fully describes how the DP algorithm compresses an original trajectory with nine points to the matched trajectory with five points.

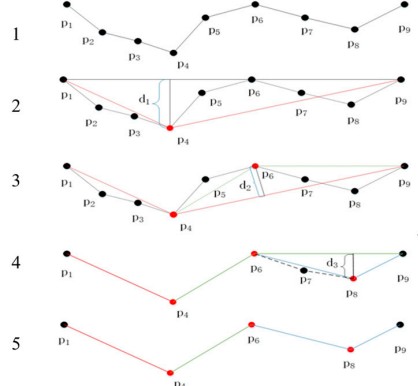

**Figure 7.** Schematic diagram of the DP algorithm.

The main steps are as follows:

Step 1: Firstly, it connects the head point and tail point of the input trajectory to form a head-tail segment. Secondly, it calculates the perpendicular Euclidean distance (PED) from the middle point (the middle point is outside the head point and tail point) to the

straight line. Finally, it finds the maximum distance, and then compares it with the given compression distance threshold;

Step 2: If $D_{\max} \leq \delta$, it discards all the middle points and only retains the head point and tail point, then the segment can be approximately viewed as the compressed trajectory of the original trajectory;

Step 3: If $D_{\max} \geq \delta$, then it keeps this point and make it the middle point to further divide the trajectory into two parts. The system repeats the above steps (step 1, step 2) for each part until all trajectories' $D_{\max} \leq \delta$ and end compression. Finally, the trajectory composed of the remaining points is the compressed trajectory.

From the above steps, it can be concluded that the compression accuracy is closely related to the distance threshold. The larger the threshold, the greater the simplification, and the more trajectory points will be reduced; On the contrary, the smaller the threshold, the smaller the simplification, and the more trajectory points will be retained, the more completely the shape of the original trajectory will be preserved. We take Figure 7 as an example to illuminate how the DP algorithm compresses a raw GPS trajectory.

Given the distance threshold $\delta$, as depicted in Figure 7, the process begins by connecting points $P_1$ and $P_9$ to form segment $\overline{P_1 P_9}$; the segment belongs to the curve. After calculating the distance $PED_{\max}$ from other points to segment $\overline{P_1 P_9}$, it can be observed that $PED(P_4) > \delta$, so the point $P_4$ should be reserved, and the segment $\overline{P_1 P_9}$ should be cut into two segments, that is, segment $\overline{P_1 P_4}$ and $\overline{P_4 P_9}$. In the segment $\overline{P_1 P_4}$, both the points $P_2$ and $P_3$ are discarded due to $PED(P_2) < \delta$ and $PED(P_3) < \delta$, and the segment $\overline{P_1 P_4}$ becomes part of the compressed trajectory. Moving to the segment $\overline{P_4 P_9}$, where $PED(P_6) > \delta$, point $P_6$ is reserved, and the segment $\overline{P_4 P_9}$ is split into two new segments: $\overline{P_4 P_6}$ and $\overline{P_6 P_9}$. In the segment $\overline{P_4 P_6}$, due to $PED(P_5) < \delta$, point $P_5$ is discarded, and the segment $\overline{P_4 P_6}$ becomes part of the compressed trajectory. In the segment $\overline{P_6 P_9}$, with $PED(P_8) > \delta$, point $P_8$ is reserved, leading to the creation of two new segments: $\overline{P_6 P_8}$ and $\overline{P_8 P_9}$. The segment $\overline{P_8 P_9}$ becomes a part of the compressed trajectory. Transitioning to the segment $\overline{P_6 P_8}$, point $P_7$ is excluded based on $PED(P_7) < \delta$. Following the aforementioned steps, only the points $P_1, P_4, P_6, P_8$, and $P_9$ remain. These points are connected to form the compressed trajectory. The compressed trajectory can be represented as follows: $TR = \{P_1, P_4, P_6, P_8, P_9\}$.

The details of the DP algorithm described in Algorithm 1.

---

**Algorithm 1:** Douglas–Peucker

---

**Input:** TR, δ. //TR is the original trajectory; δ is the distance threshold.
**Output**: resTR, // resTR is the output compressed trajectory.
```
1      procedure DPCompress (TR, δ)
2          dmax ← 0
3          index ← 0
4          resTR ← ∅
5          len ← TR.lenght − 1
6          for i = 1 → i = len do
7              d = getDistance(TR[i], TR[0], TR[len])// the perpendicular Euclidean distance
8              if d > dmax then
9                  index ← i
10                 dmax ← d
11             end if
12         end for
13         if dmax > δ then
14             leftTR = DPCompress(TR(0, index), δ)// the left part
15             rightTR = DPCompress(TR(index, len), δ)// the right part
16             resTR = merge(leftTR(0, leftTR.length − 1), rightTR(0, rightTR.length))
17         else
18             resTR = TR(0, len)
19         end if
20         return resTR
21     end procedure
```

---

### 5.2.2. SW Algorithm

Compared to the DP and TD-TR algorithms, the sliding window algorithm is one of the classical online compression techniques. Its core idea is to initiate a sliding window with an initial size of 1, starting from a specific point. Trajectory points are then sequentially added to this window, forming a segment that connects the starting point and the added point. The algorithm calculates the perpendicular Euclidean distance between the midpoint and the segment. If the perpendicular Euclidean distance of a midpoint exceeds the predefined distance threshold, the preceding point of the added point is marked for retention. The added point then becomes the new starting point for the sliding window. On the other hand, if the perpendicular Euclidean distances of all midpoints do not exceed the threshold, the algorithm continues adding new points to the window until it reaches the last point in the trajectory. We take Figure 8 as an example to illuminate how SW algorithm compresses a raw GPS trajectory.

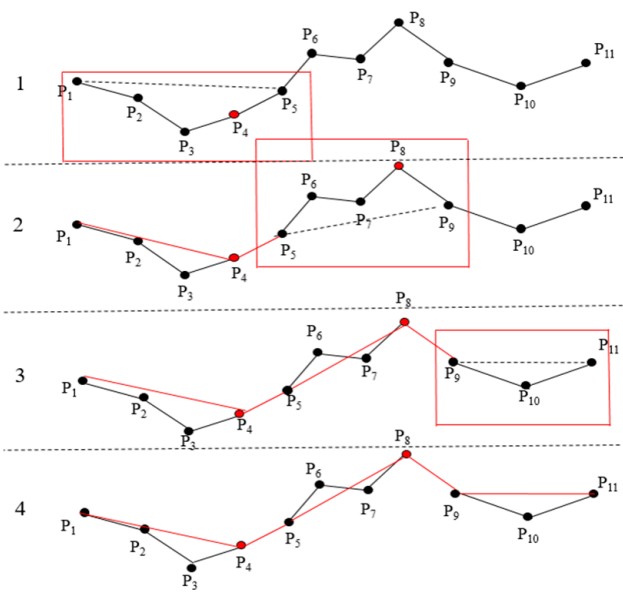

**Figure 8.** Schematic diagram of the SW algorithm.

As shown in Figure 8, given a distance threshold $\delta$, we initially set the window to size 1, utilizing point $P_1$ as the starting point for the sliding window. The points $P_2$ and $P_3$ are subsequently added, forming a new segment called $\overline{P_1 P_3}$. Since $PED(P_2) < \delta$, we continue by adding point $P_4$, creating a segment named $\overline{P_1 P_4}$. Continuously, due to $PED(P_2) < \delta$ and $PED(P_3) < \delta$, we proceed to include point $P_5$, resulting in a segment called $\overline{P_1 P_5}$. At this moment, the distance from the point $P_3$ to the segment $\overline{P_1 P_5}$ is $PED(P_3) > \delta$; we designate $P_4$ as a retained point. The point $P_5$ now serves as the fresh starting point for the sliding window, allowing us to add the points $P_6$ and $P_7$. With distance $PED(P_6) < \delta$, we introduce point $P_8$, forming a segment $\overline{P_5 P_8}$. Owing to $PED(P_6) < \delta$ and $PED(P_7) < \delta$, we persist by including point $P_9$. This creates a segment $\overline{P_5 P_9}$, and once again $P_8$ is identified as a retained point because of $PED(P_8) > \delta$. Transitioning the starting point to point $P_9$, we continue the process by incorporating points $P_{10}$ and $P_{11}$ forming a segment $\overline{P_9 P_{11}}$. As point $P_{11}$ is the final point of the original trajectory and $PED(P_{10}) < \delta$ meets the distance criterion, the traversal process concludes. By sequentially connecting the starting points of the sliding window with the retained points, the resulting compressed trajectory is obtained. Within the illustration, points $P_1$, $P_5$, and $P_9$ mark the window's starting points; points $P_4$ and $P_8$ are retained; and point $P_{11}$ represents the trajectory's termination. Thus, the compressed trajectory is derived as indicated: $TR = \{P_1, P_4, P_5, P_8, P_9, P_{11}\}$.

The details of the SW algorithm described in Algorithm 2.

---

**Algorithm 2:** Sliding-Window

---

**Input:** TR, δ. //TR is the original trajectory; δ is the distance threshold.
**Output**: resTR, // resTR is the output compressed trajectory.

1     **procedure** SWCompress (TR, δ)
2        $resTR \leftarrow \varnothing$
3        $resTR \leftarrow TR(0)$
4        $len \leftarrow TR.lenght - 1$
5        $window.size \leftarrow 1$                      // Initialize the window size to 1
6       **for** $i = 1 \rightarrow i = len$ **do**
7          $window \leftarrow window + TR(0)$
8          **if** $window.szie > 3$ **then**
9          $d = getDistance(TR[j], TR[0], TR[i])$ // 0 < j < i, calculate PED distance
10          // distance from the middle point to the head and tail point in the window
11          **if** $d > \delta$ **then** //determine the size of the distance and update the window
12          $resTR \leftarrow resTR + TR(i - 1)$
13          $window.update$
14          **end if**
15          **end if**
16       **end for**
17       $resTR \leftarrow resTR + TR(len)$
18       $return\ resTR$
19     **end procedure**

---

*5.3. Parallel Implementation Based on RDD*

Section 5.2 lists five classic trajectory compression algorithms, all of which can only be executed in a single-threaded manner and are limited to running in a single environment. In the case of massive incremental and historical trajectory data, these algorithms are insufficient to compress incremental GPS datasets with hundreds of GBs in an acceptable time cost.

This subsection will introduce how we employ Spark and RDD to perform pre-processing and compression on a large-scale trajectory dataset in a parallel way. Spark provides a programming model based on the Directed Acyclic Graph (DAG), which is more expressive than MapReduce. RDD (Resilient Distributed Datasets) is a core component in Spark ecosystem, which organizes dataset in the memory. It is a collection of elements partitioned across the nodes of the cluster that can be operated in parallel. A RDD consists of a group of partitions and each RDD depends on other RDDs. DAG maintains the linkages among different RDDs according to the source code. The job scheduler optimizes the workflow of a job and divides the job into different stages according to these linkages.

Figure 9 depicts the workflow of trajectory compression. The input dataset is GPS positioning records stored in HDFS. A dotted box stands for a RDD, and a gray square means a partition. An arrow denotes the dependency among different RDDs. All these RDDs form a pipeline to perform data pre-processing and trajectory compression. The left side lists the operators being applied on different RDDs, and the right shows the RDDs and detail elements. The primary steps are as follows:

Step 1: GPS records input. It loads the GPS record dataset from HDFS to memory and initiates the first RDD. Each GPS record contains approximately twenty fields.

Step 2: Data extraction. It extracts a group of fields from the original GPS record, which include vehicle identity, time stamp, latitude, and longitude. Each element stands for a point.

Step 3: Trajectory generation. It employs the *GroupBy* operator on the RDD generated in Step 2. In detail, the operator groups and aggregates these trajectory points according to vehicle identity, then sorts all points for each group by time, and finally generates a complete trajectory for each vehicle.

Step 4: Noise filtering. It employs the *MapValues* and a UDF (User Defined Function) operator on the RDD generated in Step 3. This UDF implements heuristic noise filtering method to generate filtered GPS trajectories.

Step 5: Trajectory compression. It employs the *MapValues* and a UDF (User Defined Function) operator on the RDD generated in Step 4. This UDF implemented a classic trajectory compression algorithm, which takes a filtered trajectory as input and outputs a compressed trajectory.

Step 6: Compressed trajectory output. It saves the RDD generated in Step 5 to HDFS.

Based on this workflow, we have implemented a pipeline-based solution that automatically performs preprocessing and compressing for continuous GPS trajectories on the Spark platform.

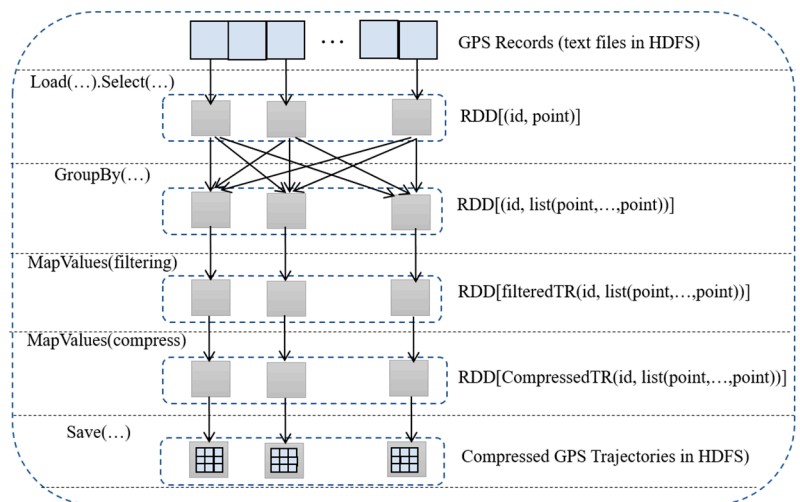

**Figure 9.** The pipeline and workflow of trajectory compression in Spark.

With the user-defined function mechanism, the end-users can accomplish a series of independent data-processing steps. For instance, all data-processing steps, such as noise filtering, map matching, data partitioning, and trajectory compression, could be implemented as user-defined functions. This capability enables operators to finely customize and control the data-processing workflow based on specific requirements.

*5.4. Evaluation Indicator*

Generally, the GPS trajectory compression algorithm can be measured from three aspects: compression time, compression ratio, and recovery effect.

**Definition 5.** *Compression time T. Compression time refers to the time required to complete compression. Assuming the timestamp of the start of compression is $T_1$ and the timestamp when compression is finished is $T_2$, then the compression time T can be defined as Equation (7):*

$$T = T_2 - T_1 \tag{7}$$

**Definition 6.** *Compression ratio. It measures the space saved by a compression algorithm. Assuming the number of an original trajectory points is $S_1$ and the number of corresponding compressed trajectory points is $S_2$, then the compression ratio R can be defined as Equation (8):*

$$R = 1 - \frac{S_2}{S_1} \tag{8}$$

*Under the premise of ensuring compression accuracy, the higher the compression rate, the better the compression effect will be.*

**Definition 7.** *Average SED error. The average SED error refers to the average error between the position of a discarded point in the original trajectory and its corresponding position in the compressed trajectory; it reflects the trajectory difference before and after compression. The calculation formula is Equation (9):*

$$\overline{SED} = \left(\sum_{i=1}^{n} SED(P_i)\right)/n \tag{9}$$

In Equation (9), n represents the number of points in the original trajectory, and SED (Pi) represents the synchronized Euclidean distance between the i-th point in the original trajectory and its corresponding point in the compressed trajectory.

## 6. Experimental Result Analysis

This section provides a comprehensive evaluation of the aforementioned methods, examining their performance in various aspects, including visualized comparison, execution time, data compression ratio, threshold, average error, and scalability. The evaluation is conducted in great detail to provide a thorough understanding of the strengths and limitations of each method.

All experiments in this paper were completed in a Spark cluster with 15 nodes. This Spark cluster consists of 1 master node and 14 worker nodes, and each node enjoys identical software and hardware configurations. Each node is a virtual machine and is equipped with 8 cores and 16 GB memory. The hardware configurations of the underlying physical machine are two 8-core Intel (R) Xeon (R) Silver 4110 CPUs @ 2.10GHz processors and 32 GB DDR memory. The software versions used are Hadoop-2.7.7, Spark-2.4.0, Scala-2.1.12, Sbt-1.2.7, etc. Five trajectory compression algorithms are implemented in Spark (Scala), and visual analysis is conducted in Python 3.6.2.

The trajectory dataset was collected from Shenzhen transportation system in January 2019. This trajectory dataset was generated by 20,000 taxis in 31 days. The number of GPS records is 1.7 billion, and the corresponding data size is 117.5 GB. Each GPS records contains many fields, such as vehicle ID, time stamp, latitude, and longitude.

### 6.1. Compression Visualization Comparison

We employ an example to conduct a visualized comparison among these five trajectory compression algorithms. The trajectory was produced by a taxi, data collected time is from 2 January 2019 12:28:53 to 2 January 2019 13:51:26, and the number of points in this trajectory is 224. According to our exploration of the trajectory dataset, the distance threshold was set to 20 m for all algorithms, and the speed threshold was set to 20 m/s for the V-DP algorithm.

Figure 10 depicts six different trajectories, the first is an original trajectory, and the other five are compressed trajectories. As aforementioned, there are 224 trajectory points in the original trajectory. As to five trajectory compression algorithms, the V-DP, SW, DP, TD-TR, and SQUISH retain 74, 82, 83, 104, and 121 points, respectively. The compression ratios of these five algorithms are 67%, 63%, 62%, 53%, and 46%, respectively. It is clearly that the V-DP algorithm has the highest compression ratio.

Comparing the two different trajectories compressed by DP and V-DP algorithms, we can clearly see a significant difference between different green circles in subfigure (b) and (c), respectively. In detail, a few points in the green circle of subfigure (c) are removed from the original trajectory because the trajectory velocity in the green circle is below the speed threshold. Thus, the V-DP achieves higher compression ratio than that of DP.

In addition, we have another interesting observation among six black circles in all subfigures. These six black circles can be classified into two categories, the first group consists of subfigure (b), (c), and (e), and the second group consists of subfigure (d) and (f). Points in each black circle of the second group are not consecutive while points in each black circle of the first group are consecutive. This is because the V-DP, DP, and SW algorithms use perpendicular Euclidean distance to calculate distances, while the TD-TR and SQUISH algorithms use synchronized Euclidean distances.

Except the above findings, it can also be observed that the red circle in the subfigure (f) is similar to the subfigure (a), while the other four diagrams do not have trajectory points of (a), this better characterizes the characteristics of the online SQUISH algorithm. Finally, after comparing the visualization of the original trajectory and five compressed trajectories, we found that all compression algorithms are largely identical with minor differences. Thus, an appropriate compression algorithm does not alter the information of the original trajectory.

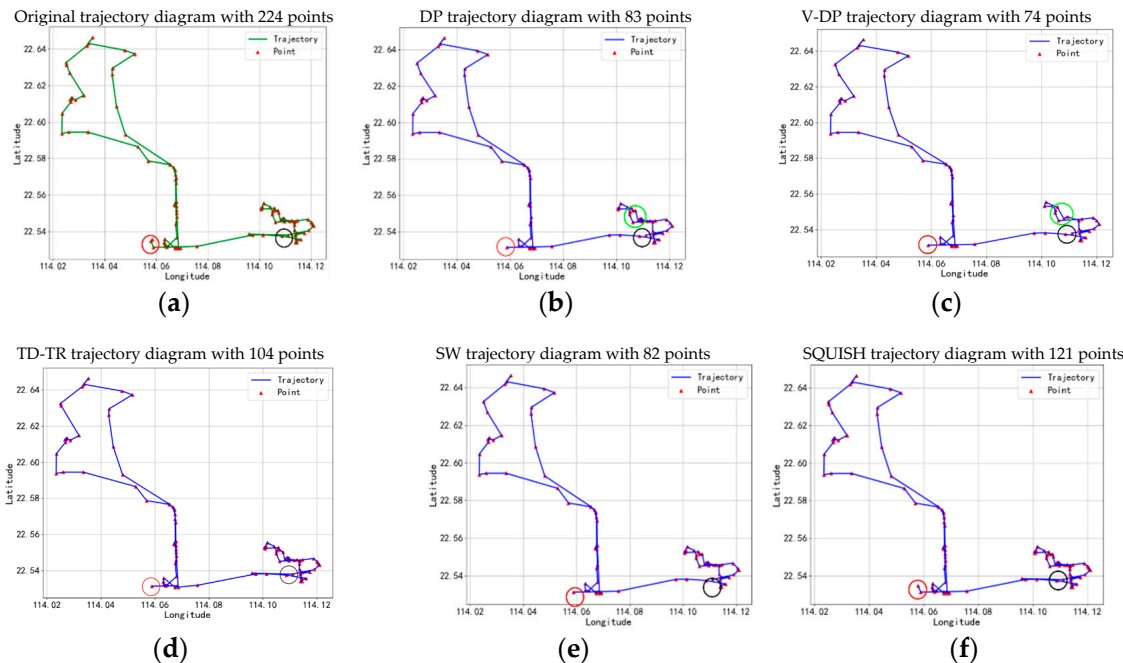

**Figure 10.** Compression visualization diagram. (**a**) Original trajectory; (**b**) DP; (**c**) V-DP; (**d**) TD-TR; (**e**) SW; (**f**) SQUISH.

*6.2. Compression Execution Time*

Figure 11 depicts the execution times of five compression algorithms on four different data sizes. These four datasets are 25%, 50%, 75%, and 100% of the original dataset size, respectively. The x-axis represents different data sizes, and the y-axis stands for the execution time. It is obvious that SW has the shortest compression times, 2.31 min, 3.88 min, 5.77 min, and 7.30 min, respectively, while TD-TR has the longest compression times, which are 8.32 min, 14.51 min, 21.18 min, and 28.35 min, respectively. From the experimental results, it can be seen that as the amount of data increases, the time of various compression algorithms increases. We can conclude that the execution time is proportional to the input size and these parallelized algorithms present good scalability.

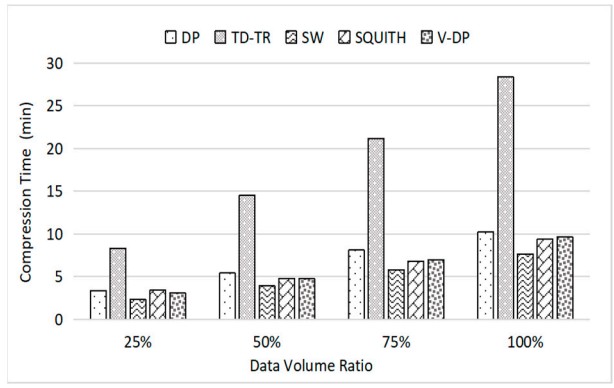

**Figure 11.** Compression times on varying data sizes.

Across four different data sets, the execution time of TD-TR algorithm is 2.65 times that of the DP algorithm on average. This is because the TD-TR algorithm is a variant of the DP algorithm, and it employs a synchronized Euclidean distance, which takes time dimension into consideration. The V-DP algorithm introduces a speed threshold to reduce the number of points in the original trajectory. Thus, the V-DP algorithm is more efficient than the DP algorithm. Specifically, the V-DP algorithm reduces the execution time by 11% compared to the DP algorithm. The time complexity of the DP algorithm is O(nlogn), where n is the number of input trajectory points. When compared to the DP algorithm, the V-DP algorithm performs a filtering strategy on the original trajectory according to the speed, reducing the number of input points. Therefore, although the time complexity of the V-DP algorithm is also O(nlogn), its actual execution time is less than that of the DP algorithm. Similarly, the time complexity of the TD-TR algorithm is also O(nlogn). It simultaneously considers the time dimension and the space dimension. In contrast, the DP algorithm only considers the space dimension. As a result, the TD-TR algorithm exhibits a significantly higher actual execution time compared to the DP algorithm.

The SW algorithm and SQUISH algorithm are online compression algorithms, the SW algorithm uses a perpendicular Euclidean distance, while the SQUISH algorithm uses a synchronized Euclidean distance, and both of them use the window method. Compared to the SW algorithm, the SQUISHs runtime is longer. In detail, the execution time of SW algorithm saves 21% time cost over the SQUISH algorithm. The time complexity of the SW algorithm is O(n), where n is the number of input trajectory points. The SW algorithm has a typical time complexity that is linear. When compared to the SW algorithm, although the time complexity of the SQUISH algorithm is also O(n), it uses a Haar wavelet-based compression technique to reduce the size of trajectory data. Thus, the actual execution time of it is much higher than that of the SW algorithm.

### 6.3. Data Compression Ratio

Figure 12 depicts the compression ratios of five compression algorithms on four different data sizes. It can be seen that the V-DP algorithm has the highest compression ratio, with an average compression ratio of 77%, while the SQUISH algorithm has the lowest compression ratio of 60%. The average compression ratio of SW is 75%, the average compression ratio of DP is 74%, and the average compression ratio of TD-TR is 64%. It means that these five algorithms can save storage costs by 23%, 40%, 25%, 26%, and 36%, respectively.

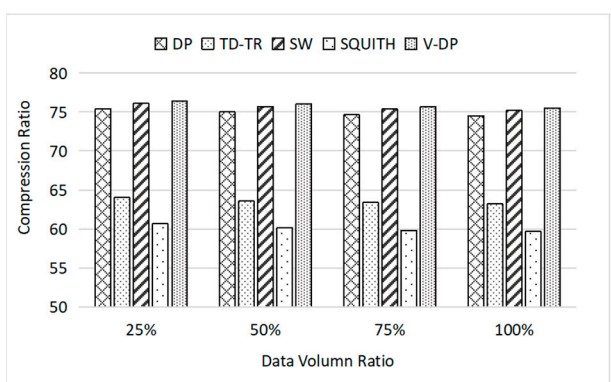

**Figure 12.** Compression ratios on varying data sizes.

In this paper, all algorithms either use a synchronized Euclidean distance or perpendicular Euclidean distance to measure the distance from a point to a segment. When using the same distance threshold, algorithms using synchronized Euclidean distances (TD-TR, SQUISH) retain more feature points than those using perpendicular Euclidean distances. Therefore, among offline algorithms, the compression ratio of the TD-TR algorithm is lower

than that of the DP algorithm and the V-DP algorithm. Among these online algorithms, the compression ratio of the SQUISH algorithm is less than that of the SW algorithm.

### 6.4. Threshold and Average Error

Considering the unlimited error of the SQUISH algorithm, this paper only compares the threshold values of the other four algorithms. By setting different threshold distances, we can conduct a systematical comparison among these four algorithms in terms of ratios and average distance error.

Figure 13 illustrates that the average error increases continuously as the threshold increases, revealing a near-proportional relationship. This is because, as the threshold increases, more points will be discarded from the original trajectory, resulting in an increase in the average error. Utilizing synchronized Euclidean distance, the TD-TR algorithm exhibits the lowest average error. With the threshold set to 5 m, the average errors of the TD-TR, DP, V-DP, and SW algorithms are 0.87 km, 1.29 km, 2.38 km, and 3.29 km, respectively. As the threshold increases from 5 to 25 m, the average errors for the TD-TR algorithm increase to 0.87 km, 2.15 km, 3.60 km, 5.20 km, and 6.95 km, respectively.

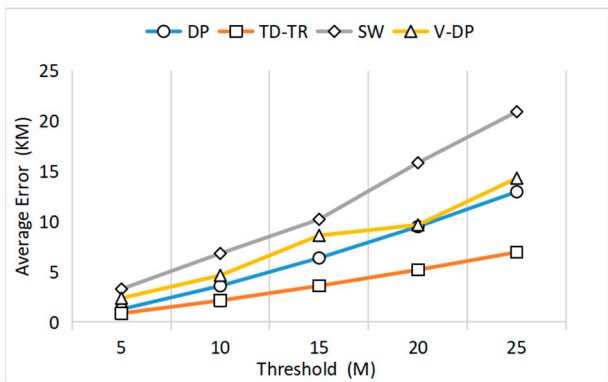

**Figure 13.** Average distance errors on varying thresholds.

Figure 14 depicts the compression ratios across varying thresholds for these four algorithms. The figure indicates that the compression ratio is only marginally affected by the distance threshold. When the threshold is set to 5 m, the compression ratios for the DP, TD-TR, SW, and V-DP algorithms are 68.18%, 63.71%, 69.54%, and 70.87%, respectively. Furthermore, When the threshold is increased fivefold to reach 25 m, the compression ratios of the DP, TD-TR, SW, and V-DP algorithms increase by only 11.22%, 8.50%, 10.12%, and 9.21%, respectively.

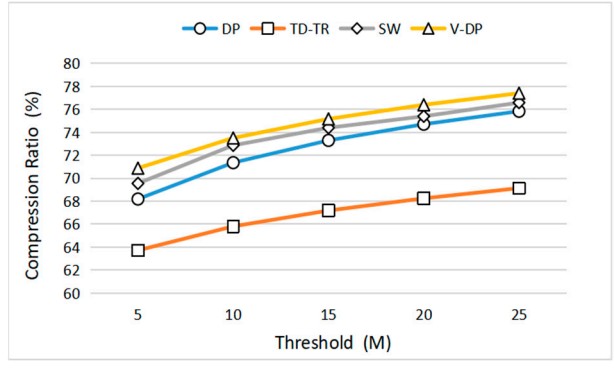

**Figure 14.** Compression ratios on varying thresholds.

### 6.5. Scalability Comparison

Figure 15 depicts the execution times for five different algorithms on four different clusters. In this experiment, we take a 27 GB trajectory as the input data and each node only

holds one executor. These four clusters include 8, 10, 12, and 14 nodes, respectively. As the number of executors continues to increase, the compression time shows a downward trend. In detail, the execution time of V-DP drops from 350 s to 169 s when the number of executors increases from 8 to 14.

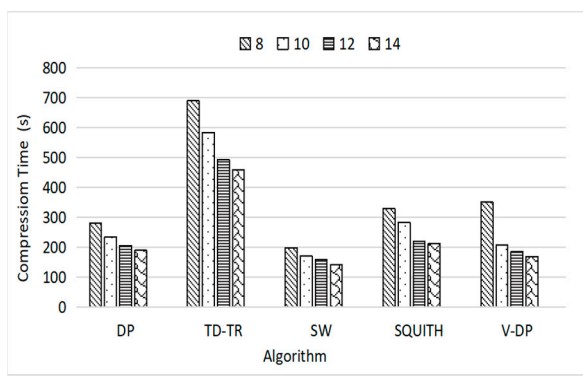

**Figure 15.** Compression times on varying nodes.

When the cluster size is 14, the average execution time among the five algorithms is 223 s. In another experiment, the V-DP algorithm takes only 438 s to compress 117.5 GB trajectory data on a Spark cluster with 14 nodes.

In these experiments, we changed the number of executors from 8 to 14 worker nodes to perform scalability comparison. Each node holds only one executor, and each executor was assigned 8 cores and 16 GB memory. According to the suggestion of the Spark development document, the number of partitions is set to two times the number of cores. For example, if the total number of cores is set to 80, then the number of partitions is 160. In our experiments, we increased the number of partitions from 80 to 240 with the step size of 80, the compression time reaches its optimum when the number of partitions is set to 160.

With regard to memory, we increased the total memory from 128 GB to 224 GB to evaluate the impact of memory size on performance. For a given dataset with a size of 27 GB and a memory size of 128 GB, the execution times of these five algorithms are 279 s, 688 s, 197 s, 328 s, and 350 s, respectively. When the memory size is increased to 224 GB, the execution times of these five algorithms decrease by 32.2%, 33.4%, 27.9%, 35.7%, and 51.7%, respectively. Figure 15 displays the execution times for each algorithm across different memory sizes.

*6.6. Query Latency Comparison*

In this subsection, a set of experiments is used to compare the performance of spatial-temporal queries before and after compression. Two spatio-temporal queries, consisting of trajectory KNN (k-nearest neighbors) and range queries, are used in these experiments.

The KNN query takes a point p, a radius r, and an integer k as inputs. It searches for candidate trajectory segments within the trajectory dataset and outputs a set of segments. Each segment in this set satisfies a condition: that the segment is located within the area determined by the center point p and the radius r. The k segments will be retained if the number of candidate segments is larger than k.

The range query takes two points and two timestamps as inputs, and outputs a set of trajectory segments. The bottom-left point and upper-right point together define a rectangular region. The two timestamps represent the start time and end time, respectively. As a result, these four input parameters collectively constitute a spatio-temporal cube. The purpose of the range query is to retrieve a set of trajectory segments from the trajectory dataset, with each segment in the set being located within the spatio-temporal cube.

In these experiments, two datasets are employed. The first dataset consists of original GPS trajectories and has a size of 117.5 GB. The second dataset comprises compressed trajectories generated using the D-VP algorithm and has a size of 27 GB. We have cho-

sen GeoMesa [34] as the spatio-temporal database to store these GPS trajectory datasets and execute the associated queries. For the sake of simplicity, we utilized GeoMesa's built-in spatio-temporal indexes. These indexes encompass XZ2 and XZ3, both designed to efficiently organize point-based spatio-temporal datasets and expedite the process of data retrieval.

Figure 16a illustrates the KNN query latencies for both the original and compressed trajectory datasets across four different data sizes. A radius of 2 km is employed for the KNN query. The query latencies for the original trajectory dataset are 5.6 s, 6.9 s, 7.3 s, and 7.6 s, respectively. In contrast, the query latencies for the compressed trajectory dataset are 3.9 s, 4.3 s, 4.5 s, and 4.8 s, respectively. In comparison to the original dataset, the compressed dataset exhibits time savings of 28.9%, 36.9%, 38.3%, and 37.2% for the respective cases. Notably, a consistent advantage of the compressed trajectory is observed as the data size reaches a certain threshold.

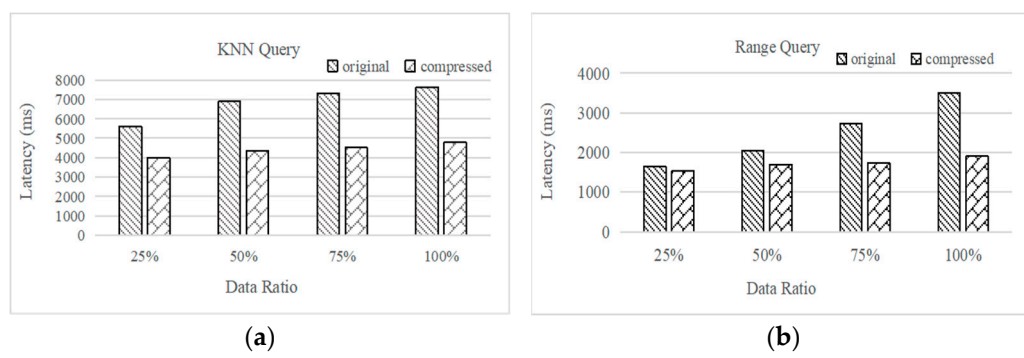

**Figure 16.** Trajectory KNN query and range query on varying data sizes; (**a**) KNN query; (**b**) range query.

Figure 16b illustrates the range query latencies for both the original and compressed trajectory datasets across four different data sizes. A 5 km × 5 km area and a time range of 5 h are defined for the range query. The query latencies for the original trajectory dataset are 5.6 s, 6.9 s, 7.3 s, and 7.6 s, respectively. In contrast, the query latencies for the compressed trajectory dataset are 1.6 s, 2.0 s, 2.7 s, and 3.5 s, respectively. When compared to the original dataset, the compressed dataset demonstrates time savings of 6.5%, 17.3%, 36.7%, and 45.6%, respectively. It is evident that this trend is observable: the larger the dataset, the greater the efficiency of the compressed trajectory.

## 7. Conclusions and Future Work

### 7.1. Conclusions

In this paper, we parallelize a set of classical trajectory compression algorithms. These algorithms consist of DP (Douglas–Peucker), TD-TR (Top-Down Time-Ratio), SW (Sliding Window), SQUISH, and V-DP (Velocity-Aware Douglas–Peucker). We comprehensively evaluate these parallelized algorithms on a very large GPS trajectory dataset, which contains 117.5 GB of a GPS trajectory dataset produced by 20,000 taxis.

The experimental results show that: (1) It takes only 438 s to compress this dataset in a Spark cluster with 14 nodes; (2) these parallelized algorithms can save 26% storage cost on average, and up to 40% storage cost; (3) the compressed trajectory can reduce the query times by 38.2% and 45.6% for the KNN query and range query, respectively.

In addition, we have designed and implemented a pipeline-based solution that automatically performs preprocessing and compression for continuous GPS trajectories on the Spark platform.

Compared with the default algorithms executed in single-threaded manner running in a single node, these parallelized algorithms can compress a large-scale trajectory dataset in an acceptable time cost and can satisfy the efficient requirement of a large-scale GPS trajectory dataset.

*7.2. Future Work*

While the current solution is capable of compressing trajectory data of several hundred gigabytes in a matter of minutes, it still fails to meet the real-time trajectory compression needs.

Generally, a streaming-oriented solution is more complex than that of batch processing mode. A typical streaming-based solution consists of two big data components: a message middleware and a stream processing engine. The former can be Kafka [35] or Pulsar [36], while the latter can be Spark Streaming or Flink [37]. For example, Kafka receives GPS positioning records transmitted from vehicles and notifies the tasks launched by Flink to retrieve these records. These long-running tasks are designed to implement trajectory compression in real-time. However, each compression task just holds a partial trajectory rather than a complete trajectory. Even worse, the points in the partial trajectory are out of order. This is the challenge we need to address in real-time trajectory compression.

The GPS positioning records are continuously generated by the running buses, and all GPS records are collected and then transmitted to the pub/sub system Kafka via different channels. In addition, a group of long-running tasks retrieves GPS records from different partitions in Kafka. The existence of multiple transmitting paths leads to data quality problems, such as incomplete data and data being out of order. For example, a set of GPS points produced by a specific bus may be received by different tasks. Similarly, the points produced by the same bus and received by the same task are in disorder. We need to design algorithms that are clever enough to solve this challenge. The potential method needs to consider three aspects: an appropriately sized time window, vehicle trajectory prediction, and candidate road segment pre-fetching.

**Author Contributions:** Conceptualization, W.X. and X.W.; methodology, W.X. and X.W.; software, H.L.; validation, H.L.; investigation, W.X.; writing—original draft preparation, H.L., W.X. and X.W.; writing—review and editing, H.L., W.X. and X.W.; visualization, H.L. and W.X. All authors have read and agreed to the published version of the manuscript.

**Funding:** This research was funded by the National Natural Science Foundation of China (NSFC) (Number: 61862066).

**Data Availability Statement:** The data presented in this study are available on request from the corresponding author.

**Conflicts of Interest:** The authors declare no conflict of interest.

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
