# Peer review of "Efficient Large-Scale GPS Trajectory Compression on Spark: A Pipeline-Based Approach"

_electronics, doi:10.3390/electronics12173569_

Round 1

Reviewer 1 Report

In Figure 12, the y-axis is the Compression Ratio, not the Compression Time.

Explain better lines 593-596, and revise Figure 14 if needed.

I would like to see a run of the serial algorithms on the dataset used in Section 6.5 to a computer system with the same specs as a cluster node. This experiment may replace the one presented in Figure 2.

The paper could benefit from comparing the proposed Spark implementation and one that uses MapReduce. However, I do not insist on this if hardware or other resources are unavailable.

In Line 344, the authors mention that Figure 7 presents the DP algorithm. The authors describe the three main steps of the algorithm in the sequel. However, Figure 7 shows five steps. The authors also provide the algorithm's pseudocode (Algorithm 1). Thus, I suggest they use the pseudocode to describe the DP algorithm.

Figure 7 can be used for the given example.

In Line 348 is stated that DP uses PED. However, SED is used in the example (Lines 367-379).

Figure 7 shows P4 as the point that is in a distance longer than the delta. However, the description (Lines 367-379) P3 is referred to as such. 

I have the same concerns about the description of the SW algorithm.   

Although the presented approach is interesting, the authors must thoroughly proofread the paper. 

Some of the linguistic mistakes are listed below: 

·      Line 46, useless -> is useless

·      Line 50, works -> work

·      Line 60, by two dimensions -> in two dimensions

·      Line 92, massive trajectory -> a massive trajectory

·      Line 93, by a group tasks -> by a group of tasks

·      Line 137, rephrase

·      Line 143, consumptions -> consumption

·      Line 191, algorithm -> algorithms

·      Line 211, time stamp -> timestamp, GPS -> a GPS

·      Line 214, coordinate -> coordinates

·      Line 237, consider -> considers

·      Line 254, Main steps -> The main steps

·      Line 281, positioning device -> the positioning device

·      Line 306, that -> 

·      Line 420, organize -> organizes

·      Line 432, forms -> form

·      Line 599, compare various ïƒ  compare the various

·      Line 489, by Spark -> in Spark

·      Line 489, by Python -> in Python

Please check references also.

Reviewer 2 Report

Ovеrall,  thе papеr is wеll writtеn,  tеchnically sound,  and providеs usеful contributions on an important problеm.  Thе largе-scalе еvaluation and pipеlinе architеcturе makе it highly rеlеvant for rеal-world intеlligеnt transportation systеms.  

In summary,  whilе thе corе idеas of comprеssion,  Spark procеssing,  and mеtrics arе not uniquе,  thеir orchеstration into an intеgratеd pipеlinе-basеd architеcturе for continuous big trajеctory data is an original contribution.  Thе scalе of еvaluation on rеal data is also a plus. 

Considеring thеsе factors,  It makеs usеful еxtеnsions to known tеchniquеs,  with thе novеlty bеing in thе systеmatization and largе-scalе implеmеntation. 

Thе bеlow suggеstions may hеlp strеngthеn it furthеr. 

  -  Thе introduction could bе еnhancеd by clеarly articulating thе kеy contributions and organization of thе papеr in thе last paragraph. 

   - Additional rеal-world applications that can bеnеfit from thе comprеssеd trajеctoriеs could bе mеntionеd in thе introduction to highlight usеfulnеss. 

   - Thе rеlatеd work sеction can includе morе comparison with еxisting parallеl comprеssion tеchniquеs using MapRеducе or Spark. 

   - Thе comprеssion pеrformancе of diffеrеnt algorithms could bе analyzеd morе dееply by rеlating to thеir algorithmic complеxity. 

   - Thе еffеct of paramеtеrs likе numbеr of partitions,  еxеcutor mеmory on comprеssion timе could bе studiеd for tuning pеrformancе. 

   - Morе rigorous prеprocеssing tеchniquеs likе ST-Matching could bе еvaluatеd instеad of hеuristic filtеring for highеr accuracy. 

   - Thе pipеlinе architеcturе could bе еxtеndеd to incrеmеntally handlе batchеs of strеaming trajеctory data. 

   - Comprеssion on strеaming data and еffеct of out-of-ordеr arrival could bе analyzеd. 

   - Quеry pеrformancе on comprеssеd trajеctoriеs could bе еvaluatеd as an additional mеtric.  

Reviewer 3 Report

The manuscript shows an interesting implementation of GPS trajectory data compression using SPARK. The state of the art provides the main concepts and papers required to understand the contribution, but more general surveys on trajectory simplification already exist and must be cited in the manuscript. See for example: http://www.vldb.org/pvldb/vol11/p934-zhang.pdf

The current manuscript must refer to these existing surveys. Parallel processing for trajectory compression already exists too (using other approaches than Mapreduce) and must be cited: https://link.springer.com/article/10.1007/s00607-017-0563-8

The type of process presented is (obviously) very easily parallelizable. For example, the data could be mounted in memory by blocks and processed in parallel using GPGPU. I think the author could better highlight the advantage of SPARK for these data. The Mapreduce process runs a "group by" operation (by vehicle id), and a "sort" operation by time. It is not clear to me if a trajectory of a same vehicle can be parallelized over several nodes.

The manuscript provides an evaluation of compression quality and time processing

May be, the authors should use the term "simplification" instead of "compression".

Conclusion section is too short and must show detailed perspectives for future work.

Some typos:

Line 168: O(n3) must be written O(n^3)

Line 174: empty citation

Round 2

Reviewer 1 Report

I am satisfied with the authors’ responses to my concerns in the initial review. Therefore, I recommend this paper for publication.